# The impact of technical efficiency on firms' value: The case of the halal food and beverage industry in selected countries

**Sylva Alif Rusmita**[1], **Siti Zulaikha**[1]*, **Nur Syazwani Mazlan**[2], **Nuradli Ridzwan Shah Bin Mohd Dali**[3], **Eko Fajar Cahyono**[1], **Indria Ramadhani**[4]

**1** Department of Sharia Economics, Faculty of Economics and Business, Airlangga University, Surabaya, Jawa Timur, Indonesia, **2** Department of Economics, Faculty of Economics and Management, Universiti Putra Malaysia, Serdang, Selangor, Malaysia, **3** Department Business and Management, Faculty of Economics and Muamalat, Universiti Sains Islam Malaysia, Bandar Baru Nilai, Negeri Sembilan, Malaysia, **4** School of Graduate and Professional Studies, INCEIF University, Kuala Lumpur, Malaysia

* siti-z@feb.unair.ac.id

**Data Availability Statement:** All relevant data are within the paper and its Supporting Information files.

## Abstract

The market for the halal food and beverage industry sector has experienced rapid growth in recent years, which indicate excellent investment opportunities. This paper examine the effect of Technical Efficiency (TE) on firm value in 5 selected influential countries in halal food and beverage sector based on Global Islamic Economy Report 2020. Two steps estimation was used to run the data, using the Stochastic Frontier Analysis (SFA) model to determine the company's TE and panel data to test the effect of TE through firm value. The results show that Indonesia has the highest score for TE (62%), followed by Pakistan (59%), South Africa (57%), Malaysia (55%), and Singapore (52%), which means, in general, there is inefficiency in allocating resources over 38% up to 48% and needs to be improved by halal food and beverage companies in. Regarding panel data, all countries sample except Pakistan highlight that TE significantly affect company value. It indicates that the crucial part of managing efficiency can be a sign in stock market performance. The result shows that company managers should set efficiency strategies to their business process for creating sustainability and increase their value in the capital market. As for investors, this TE can be used as an indicator before choosing company stocks; if the company is efficient, then it is worthy of being one of the portfolio assets. Form the government side, the finding can help them to set appropriate policy setting to boost halal food and beverages industry such as giving subsidy or incentive to increase the efficiency ability of halal food and beverage companies and identify the industry's strength by comparing the result of TE between 5 countries.

## Introduction

Food and drink are basic human needs. For Muslims, there are specific rules for consuming these; in other words, both food and drink must be halal in substance. The number of Muslims in the world as of 2019 was 1.9 billion, equivalent to 24% of the world's population [1]. The

**Funding:** The author who is received the research grand is Sylva Alif Rusmita from Airlangga University and the letter agreement is 929/UN3/2021 (https://drive.google.com/file/d/1FhoMKQaiieuG1XHbxpFXoy9RbsGgL4sd/view). The funders role are design, analysis, and decided the publisher.

**Competing interests:** The authors have declared that no competing interests exist.

magnitude of this opportunity is predicted to have an impact on the growth rate of the halal food and beverage market, which will increase by 6.3% from 2018–2024 equivalent to 1.972 billion USD [2]. This is good news for halal food and beverage industry players, including for countries that are ranked in the top tier in the halal food sector.

In 2019, 35% of Muslims' income, which is equivalent to 1.17 trillion USD, was allocated to meet their needs for halal food and beverages. In 2024 this is predicted to increase by 1.38 trillion USD [2]. In several countries with a majority Muslim population, such as Indonesia, support for this sector is given through government regulations and targets developed to improve the performance of the halal food and beverage sector, through halal sector master plan policies [3]. The existence of good macroeconomic factors in the form of global market shares, as well as encouragement from the government in several countries, either through policies or injections of investment funds, can encourage the fast development of the halal food and beverage sector. This aligns with GIE's (Global Islamic Economy Report) 2020 prediction of an increase in the consumption of halal food and beverages from 2019 to 2024 by 21%.

Good macroeconomic factors can be in vain if the company does not know its performance internally. The tools to measure the company's performance is to calculate its efficiency level. Efficiency is a measurement of effectiveness in order to minimize wasted time, effort and skills in production processes [4]. One way to measure efficiency is by using the SFA (Stochastic Frontier Analysis) technical efficiency method, which identifies the company's efficiency and inefficiency values [5].

The increase in the size of the halal food and beverage sector also has an impact in terms of investment. The GIE Report 2020 data revealed that the largest investment opportunity is in the halal food sector at 51% compared to other sectors. Muslims aim for future prosperity and must pay attention to the halal element in their investment products. A good investment apart from the halal element, can also be seen from the profits related to the company's performance. Better performance means the company's income will increase and provide good returns, which will improve the welfare of investors, as reflected in the large increase in?? the company's value [6].

Research on technical efficiency in the food and beverage industry has been widely conducted using DEA (Data Envelopment Analysis) methods in various countries [7–12]. The study fills a gap in the literature by measuring technical efficiency using the SFA method with samples of halal food and beverage companies. The selection of SFA models, although in accordance with economic-based research [13], is rarely used in Islamic capital market case studies [14].

Previous research on technical efficiency and its impact on corporate value has been conduct and shows that technical efficiency has a significant positive impact on the value of electronic companies in Japan, of technology companies in the USA, and of non-finance companies in Australia, which indicate that the technical efficiency score significantly influences the company's value [15–17]. So far, there has been no research focusing on samples of halal food and beverage companies, so it can be concluded that the novelty of this study lies in a sample of the research that is five influential halal food and beverage companies based on version of the State of the Global Islamic Economic Report 2020 and has Islamic capital market index during research period, namely Malaysia, Singapore, Indonesia, South Africa, and Pakistan as an individual analysis was performed for each sample of the country, so that this research sample has a global scope.

Companies in the food and beverage sub-sector are part of consumer goods stocks that are considered defensive in conditions of economic recession, due to the product part of human basic needs [18]. In 2020 there was an economic recession due to the Covid-19 pandemic, and halal food and beverage, which are listed on the stock market, showed their ability to maintain their business performance during that time. Thus, to make a company sustainable, managing

its production efficiency could be one of the solutions, boosting the company's performance in the real sector and the capital market. One of the tools to calculate firm performances is Tobin's Q, which can identify managerial ability and considers investment growth [16, 19, 20]. Efficiency results for economic conditions before and during the pandemic can be compared to illustrate how significant the impact of the pandemic was in a country, especially in the halal food and beverage sector. Moreover, this study will also analyze the effect of efficiency on Tobin's Q company performance for each country.

This study will discuss how the technical efficiency of halal food and beverage companies in the five biggest influential countries (according to Global Islamic Economy Report 2020) affect and impact company value. Considering the large market share and investment opportunities in the world's halal food and beverage industry, the contribution of this research, in addition to filling previous research gaps, will also be helpful for halal industry companies so they can know their production performance and see its impact on the company's value so that in the future the company can continue to develop efficient use of production resources. For investors in these companies, the results of the efficiency scores can be used as a reference to assess the company's performance.

The results of this study will be helpful for all parties concerned in the halal food and beverage industry. First, halal food and beverage companies can know their operational performance, and companies with reasonable efficiency levels, can serve as benchmarks for other similar companies. In addition, the company can find out how the investor's perspective is related to the level of efficiency, so it can make the right managerial decisions to form a good corporate image in the eyes of investors through its production management. Second, for investors, efficiency scores can be used as one of the considerations when investing in the halal food and beverage sector. In addition, investors can identify investment potential, especially for Muslim investors. This study's results can guide investment in halal food and beverage sector companies that meet sharia compliance. From the government side, this research can provide input related to regulation of the capital market in the halal food and beverage sector. Furthermore, it can be an input for governments to encourage the domestic halal industry, considering that based on GIE Report 2020 data, this sector has the most important investment opportunities. Fourth, for the field of science, this study presents the effect of TE (Technical Efficiency) on firm value by paying attention to microeconomics and macroeconomics as control variables and filling the gaps from previous research so that it can be a new reference for future study, especially in research related to the halal industry.

The writing of this research is structured as follows. The introduction contains the background for selecting themes and research topics, the phenomena and data surrounding them, and the research objectives. Furthermore, the literature review is separated into three sections which are an explanation of the theoretical basis for determining the hypothesis, technical efficiency explanation, empirical review, which explains the empirical findings of previous research on topics relevant to this study, and it concludes with a hypothesis. After that, this study presents data and statistical methods that describe the types and sources of data and these are followed by an empirical framework that provides an empirical model for this study. In the end, there is a result and discussion section to analyze the essential findings of the research, and this study ends with a conclusion and policy implications.

## Literature review

### Signaling theory

Signal theory explains how information about a company's performance can provide an overview for stakeholders [21]. In the case of the stock market, signal theory can encourage how

investors behave toward a company's stock. There are three elements in this theory: signaler, signal, and receiver [22]. The company can be categorized as a signaler because it can inform the state of the company through its financial statements. Shareholders can capture the existence of various pieces of information, or signals, as the receiver. There are positive and negative signals that shareholders can receive. Signals can erode the gap between what the company knows and what investors want to know. When investors receive specific signals, there are various response options, depending on the type of investment and investors' ability to understand existing company information [21, 22].

Positive and negative signals explain the company's performance and depend on how the company uses capital market transactions to communicate the quality of its performance and efficiency to investors. From the company's operational point of view, efficiency in the production process can improve machine or human power performance, and optimize the use of raw materials [5]. This means that efficiency can have an impact on the success of the company, especially in increasing the amount of production and revenue. Operational performance is reflected in profitability ratios such as operating income ratio, ROI, and ROE. Investors generally consider these ratios in investing and they are used by companies to provide signals to investors [23]. Investors will place their funds in stocks with a high company value because doing so can increase their welfare or provide a large investment return [15, 17, 24]. This indicates how important it is for the company to maintain efficient performance because it can have an impact on profitability, which will affect the value of the company.

## Technical efficiency

In economics, technical efficiency refers to a production process's ability to create the maximum potential output, given a set of inputs. A company's technical efficiency is often determined by comparing its actual output to the greatest output that may be achieved given the same inputs and technologies. This notion is crucial for understanding the performance of enterprises and industries, and it has received a lot of attention in the economics literature [5]. Efficiency occurs when the input-output has reached a certain point and cannot be increased without worsening productivity so that efficiency can be used as a tool for evaluating the performance of production activity units [25]. One of the tools to measure efficiency is the technical efficiency (TE) method that uses a score from 0–1. It reflects the company's ability to maximize output levels with optimal use of inputs, and the closer to 1 the score, the better the level of efficiency and vice versa [5].

There are two types of TE usually used; the first is non-parametric DEA (Data Envelopment Analysis), the second, parametric SFA (Stochastic Frontier Analysis). The DEA approach has been widely used to determine TE in the food and beverage sector in several countries [8–12, 26–30] and known as nonparametric method for assessing the efficiency of a collection of operating units used the weighted for each variable [31]. However, choosing input and output variables is less specific and can result in multicollinear variables. Even in this model, it is possible to use two output variables. In addition, the models does not account for the error term and cannot test hypotheses on parameters. Unlike SFA, efficiency scores cannot be calculated for all observations.

On the other hand, the SFA parametric approach is used to calculate efficiency, especially in the Islamic capital market sector [14]. The SFA method is one method for measuring technical efficiency. It is commonly utilized in empirical studies of company performance because it allows for the measurement of both technical efficiency and inefficiency [31]. The method entails estimating a production function with both random and systematic components. The unobservable factors that affect the production process, such as weather or other external

factors, are represented by the random component, while the systematic component indicates the firm's level of technical efficiency.

The SFA approach has several advantages over DEA methods of measuring technical efficiency, including the ability to differentiate between technical efficiency and inefficiency, the ability to control for external factors that affect production, and the flexibility to deal with different functional forms and distributional assumptions [5]. As a result, it has become a common method for assessing technical efficiency in a variety of economic sectors, including agriculture, manufacturing, and banking. As well, it is able to be run with two step estimations [32–35].

The SFA is predicated on the functional form of production functions [36]. This approach can be used effectively on data with measurement error, whereas the DEA model makes no assumptions on the functional form of the model and hence cannot properly cope with the presence of measurement error in the data. In addition to permitting the selection of the optimal functional form, stochastic frontier analysis also incorporates random error and statistical noise. Scores of efficiency measurement are also attainable with the parametric method, which yields more precise findings for efficiencies [37].

Overall, technical efficiency is a key issue in economics that has been thoroughly researched using a variety of empirical methodologies. The SFA technique is a popular method for measuring technical efficiency that has various advantages over other methods and has been utilized in several business and industry performance evaluations. Moreover, as this study uses panel data with input from a lot of variables and output from one variable, SFAs are appropriate tools to calculate technical efficiency rather than DEA because with SFA we can get specific output with all the compatibility that was discussed above. In the measurement of TE, there are provisions for the input-output selection approach. This study uses a production approach, referring to the research of [38–42]. The selection of input variable in this study includes current assets, fixed assets, liabilities, and capital. Operating income is used as an output variable. The selection of these variables is considered appropriate because they can reflect efficiency through a production approach [38].

## Empirical review efficiency impact on firm value

The halal food and beverage sector is rapidly growing, especially in Muslim countries, such as Indonesia and Pakistan [43–46]. Consequently, many food and beverages companies certify their products to enter this industry, which can be seen from the research of [47–49], which explains how this industry has a domino effect on other industries like halal tourism. Halal certification of food and beverages also impacts the business growth to expand the target market. Still, several things should be considered related to supply chain cost, certification cost, maintaining food production and quality, and building awareness when Muslims are not a majority community [47]. As a result, to get more equity for make the business sustainable, halal food and beverage companies can choosing the option by listing in stock market The stock market is a liquidity market for the firm to get funding. However, the firm becomes more public after listing there, and every investor can join the company as a shareholder. It means the company's performance is not only noticed by internal staff, [the firm's managers] but also by external stakeholders [investors] because the firm's performance can impact profitability as well as investor wealth [50].

One of the crucial tasks for the company is to allocate their resources efficiently and transform them into output, which can be measured as technical efficiency. Several kinds of research have been conducted to indicate the relationship between technical efficiency and firm value, the result describes consistency finding that technical efficiency score can be

increase by adding firm asset, equity, liability, and employee in different business sectors and countries [8–12, 15, 17, 24, 26–30, 33, 34, 37, 51]. For instance, research from [15] found a significant positive impact from technical efficiency on firm value in Japanese electronics companies. On the other hand [17] explains a similar pattern in 5 high-brand IT companies in the USA over the 2001–2015 period. Both [15, 17] emphasize how technical efficiency using the SFA production approach is the fundamental issue to consider when increasing firm performance from the perspective of the company and investors. The other finding from [16], whose paper used leverage, investment opportunity, fixed asset, operating profit, and firm profitability as a proxy for input variables and average stock return as output variable, shows all the input variables positively significantly impact technical efficiency score using the SFA method in Australian non-financial companies which were listed on the ASX during 1995–2013.

The two steps method of SFA has been widely used in other research. Such as combines the SFA method and GMM (Gaussian Mixture Model) to identify the impact of institutional quality on energy efficiency in BRICS (Brazil, Russia, India, China, and South Africa) countries [34]. The same method is also applied by [33] in using a South Asian sample to measure the impact of technology innovation on energy efficiency. The other paper used SFA and ARDL to investigate the long-term and short-term impact of energy efficiency on the carbon footprint in the ASEAN region [51]. While research from [37] compares two SFA approaches using a technical efficiency approach, the other uses environmental efficiency. The four papers emphasize the critical scope of efficiency in energy and environmental research with several different two steps and approaches, means the two steps method of SFA can be combine with other method, and this paper try to expand the scope by combining it with panel regression

The results of previous studies in different industrial sectors consistently show that firm value has a significant positive effect on the SFA technical efficiency score, with the dependent variable Tobin's Q as a representation from the investor's perspective [15, 17, 24, 52, 53]. Other research by [52, 53] also showed similar results, even though they used an efficiency ratio approach and enterprise value as a proxy for company value. In short, from the previous papers mentioned above, the research halal food and beverage sector has been widely discussed using the DEA approach [8–12, 26–30] in several different countries, or the SFA approach [15, 17, 24, 33, 34, 37, 51]. Unfortunately, research focusing on halal food and beverages is limited. The current research mainly focuses on knowing tourism sector which is impacted by the halal food and beverages, halal supply chain, and halal awareness [44, 45, 47, 48, 54–56]. Therefore, this article uses a sample of halal Food and beverages because this sector has massive growth globally and discusses the firm managerial perspective, which is measured by technical efficiency. At the same time, it looks at how investors react to firm performance from a capital market perspective.

## Hypothesis

The signaling theory explains how every single activity internal to the firm can be a sign for investors to make investment decision. The sign can be positive or negative, depending on investor behavior and preference. Technical efficiency [TE], is one of the cues which can be used by investors to decide on their investment. Furthermore, previous studies from [15, 17, 24, 52, 53] revealed that technical efficiency has a positive influence on firm value. A good level of efficiency describes the company's operations using its resources to produce output that runs optimally so that there are no idle resources and to produce maximum output [36]. Efficient companies can increase their operating income and have an impact on the welfare of investors, which can be measured by the ratio of company value which is the perception of investors [19]. Thus, the hypothesis formed in this study is:

H1: The higher the technical efficiency value, the higher the firm value

## Methodology

This study uses a quantitative approach to analyze numerical data using mathematical models that aim to develop theories and hypotheses from the investigated phenomena [57]. The data sources used were OSIRIS Database and investing.com from five selected halal food and beverage companies, based on the Global Islamic Economy Report 2020, which has an Islamic capital market. In addition, cross-checking from the annual financial statement was used to increase the validity of each company's data. The population of the study was all food and beverage companies located in the top 5 countries with an Islamic Index, and which were consistently operating between 2017 until 2021. The countries concerned are: Malaysia, Indonesia, Singapore, Pakistan, and South Africa. Purposive sampling techniques were carried out to determine the study sample, which consisted of 51 firms, as indicated in Table 1.

The research method used is two step estimations referring to previous research [15–17, 33, 34, 51]. The first stage uses the SFA method to measure halal food and beverage companies technical efficiency score by using Frontier 4.1 software. Once the results are known, the technical efficiency score will be used as an independent variable in the second stage. Eviews 9 software then used to run panel data regression method to determine the technical efficiency effect on the firm's value as its dependent variables.

### Stochastic frontier analysis (Step 1)

This research uses the SFA parametric approach to measure technical efficiency because it is suitable for use in the field of Economics [13] and it also fills the gap in the SFA literature, which is rarely used in capital market research [14]. Mathematically, the technical efficiency equation with the Cobb-Douglas SFA function is as follows:

Stochastic Frontier Analysis

$$lnq_i = \beta_0 + \beta_1 lnx_i + v_i - u_i \qquad (1)$$

$$\text{or} \quad q_i = \exp[\beta_0 + \beta_1 lnx_i + v_i - u_i] \qquad (2)$$

$$\text{or} \quad q_i = \exp(\beta_0 + \beta_1 lnx_i) \times \exp(v_i) \times \exp[-u_i] \qquad (3)$$

Because this research is included in the panel data model, the SFA model is written in Eq 4 so that this research model is mathematically described in Eq 5 [5].

$$\ln q_{it} = x_{it}^t \beta + v_{it} - u_{it} \qquad (4)$$

**Table 1. List of sample.**

| No | Country | Composite Shariah Index | Halal Food & Beverage Companies |
|---|---|---|---|
| 1 | Malaysia | FTSE Emas Sharia Index | 17 |
| 2 | Indonesia | ISSI | 12 |
| 3 | Singapore | FTSE Singapore Sharia Index | 8 |
| 4 | Pakistan | PSX-KMI | 7 |
| 5 | South Africa | JSE Sharia | 8 |
| | **Total** | | 51 |

Source: data processed

$$lnq_{it} = \beta_0 + \beta_1 lnCA_{it} + \beta_2 lnFA_{it} + \beta_3 lnLi_{it} + \beta_4 lnE_{it} - u_{it} + v_{it} \qquad (5)$$

Description:

$lnq_{it}$ = Operating Income [output variable]

$\beta_1 lnCA_{it}$ = Current Asset [input variable]

$\beta_2 lnFA_{it}$ = Fixed Asset [input variable]

$\beta_3 lnLi_{it}$ = Liabilities [input variable]

$\beta_4 lnE_{it}$ = Equity [input variable]

$u_{it}$ = Inefficiency

$v_{it}$ = Statistic noise

## Operating income

The output variable used in this study is operating income. Operating income is the difference between sales revenue and production operating expenses, such as COGS, sales expenses, general expenses, and administrative expenses. [11, 38, 41] state that operational income can explain the amount of company production or output, and can reflect the output the company generates during its production process.

## Current assets

Current assets are all short-term assets owned by the company, where they are quickly converted into money in less than 12 months. Those classified as current assets include cash and cash equivalents, receivables, short-term investments, inventory, and prepaid expenses. Previous research results showed that higher current assets owned by the company impact the level of technical efficiency of the company, which can be seen from the t-ratio: if it is higher than the p-value (5%), it means the current asset gives significant effect through firm efficiency [10, 12, 28, 38, 39, 41, 42].

## Fixed assets

Fixed assets are long-term corporate wealth that convert assets into money over 12 months. Fixed assets include vehicles, production machinery, land, intangible assets, and long-term investments. The use of fixed assets as input variables can indicate the level of productivity of the assets owned by the company, such as the use of machinery and buildings against the number of goods produced. The higher the value of fixed assets (t-value higher than p-value at 5%), the higher the technical efficiency of the company [9, 12, 28, 38, 39, 41].

## Liabilities

Liabilities describe the overall long-term debt that has a repayment maturity of more than 12 months, and also short-term debt that has a repayment maturity of less than 12 months. This study uses the total liability variable because liabilities are often considered levers for companies to increase their productivity. In addition, liabilities have a significant positive effect on the technical efficiency of the company. A higher t-ratio of liabilities than the p-value of 5%, indicates a big impact toward firm efficiency [8, 11, 38].

## Equity

The company's capital or equity comes from two primary sources. First, the paid-up or contribution capital is the amount of cash or other assets that shareholders pay to the company. The

second comes from the retained earnings, which is the company's net income, which is then reinvested for the company. Other sources classified as capital are treasury stocks and other comprehensive profit accumulations. Capital has a positive effect on technical efficiency, where when capital increases, the company becomes more efficient, as can be seen through the t-ratio: if it is bigger than t-value (5%) the variable is extremely significant [10, 28, 38, 40, 42].

The selection of variables to measure technical efficiency includes current assets, fixed assets, capital, and liabilities as input variables, while operating income is output. This selection refers to previous research, which explained that the variable is appropriate for the production approach that is the main activity of food and beverage processing companies [38–40, 42]. The result of TE is from 0 to1. The closest to 1 indicates an excellent efficiency score and vice versa.

## Panel data regression (Step 2)

The next step from SFA is to use panel data regression to determine the effect of efficiency on firm value. This research method has the advantage that it does not require a classical assumption test and is an appropriate method for analyzing the relationship between independent variables and their dependents [57]. The fit model for panel regression can be known by doing several tests. There are tests that should be done: Chow, Hausman, and Lagrange Multiplier. These test will indicate the best model for panel regression, either FEM (Fixed Effect Model), REM (Random Effect Model), or CEM (Command Effect Model). The Chow test considers CEM or FEM, whereas the Hausman test is used to know the better option, either FEM or REM; and the Lagrange Multiplier test can be decided between CEM or REM [58].

The dependent variable is the firm value, calculated using Tobin's Q (TQ) ratio. The use of TQ is considerd because of stock price movements, management's ability to manage resources, and the possibility of investment growth [16, 19, 20]. The SFA TE score was used as the independent variable. Because the number of independent variables is only 1, this study includes control variables which are firm size, leverage, capital, growth, and age, to avoid bias in the research model [15, 17, 53, 59]. The panel data regression model can be formulated as follows:

$$Q_{it} = \beta_0 + \beta_1 TE_{it} + \beta_2 FS_{it} + \beta_3 L_{it} + \beta_4 G_{it} + \beta_5 I_{it} + \varepsilon_{it} \tag{6}$$

$Q_{it}$ = Tobin's Q
$\beta_1 TE_{it}$ = Technical Efficiency
$\beta_2 FS_{it}$ = Firm Size
$\beta_3 L_{it}$ = Leverage
$\beta_4 G_{it}$ = Growth
$\beta_5 I_{it}$ = Inflation
$\varepsilon_{it}$ = Statistic noise

## Firm value

The public listed companies have the aim of maximizing the outstanding present value per share or what is commonly called the firm value [19]. It is because the high firm value indicates the company is trying to manage the resources it has now to gain higher future profit. Furthermore, the high firm value is a sign that the company can taking into account the risk, focuses on managing cash flow rather than just profit, and taking into account social responsibility. With all the activities mention before, it can affect stock prices. Firm value is related to stock prices, because stock prices reflect investors' perceptions of certain companies, where investors are ready to pay a certain amount for each share [19]. One indicator used to measure the stock price method is Tobin's Q because it is able to reflect the company's operations and investments in generating added value. Tobin's Q value is obtained from the company's financial

statements through the following formula:

$$Tobin's\ Q = \frac{Market\ Value\ of\ Equity + Book\ Value\ of\ Liability}{Book\ Value\ of\ Total\ Asset} \tag{7}$$

### Technical efficiency

The efficiency score is ratio data obtained from the result of the SFA process in the first step. The efficiency scores range from 0–1, the closer to 1 then the more efficient the company [5]. To obtain this score the SFA method described in Eq 5 is used. Previous research has shown a significant positive relationship between efficiency scores and firm value [15, 17, 24, 52, 53].

Although the two steps (SFA and panel data regression) both see a cause-and-effect relationship and are included in the panel data, there are differences where, in the SFA method, the equation model used is MLE [5], while in the panel data regression, OLS is used. Consequently, a test is performed to select the best model (FEM, REM, or CEM) [57]. In short, the SFA model has a final score representing the company's efficiency level from step 1. This score will then be used as an independent variable in step 2. The connection of each variable is captured in the Fig 1.

Step 1

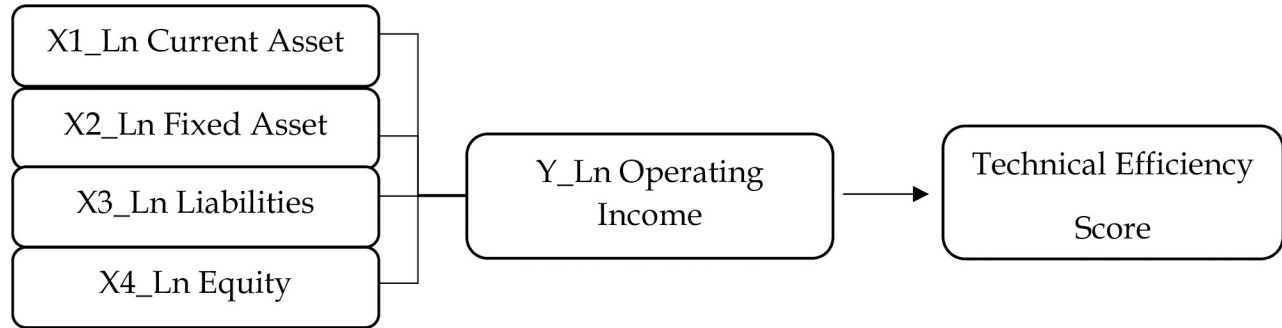

Step 2

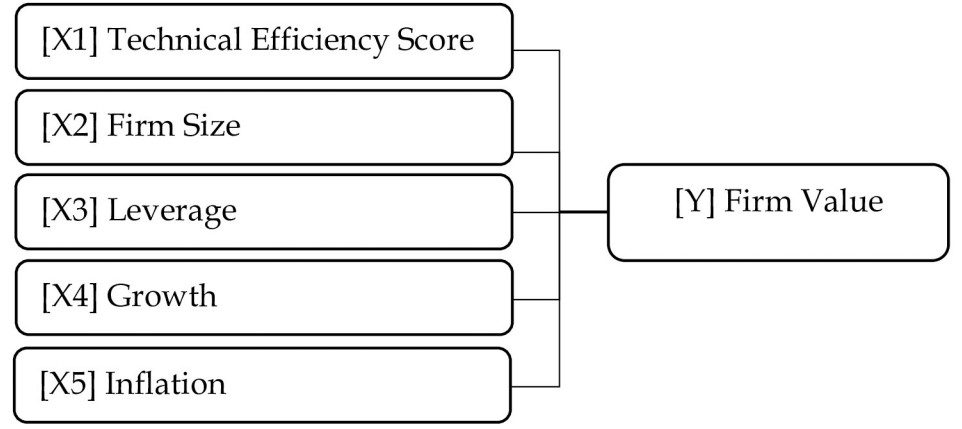

**Fig 1. Conceptual framework.**

## Result and discussion

### Statistical result

**Descriptive statistic.** Table 2 provides summary statistics of data for the stochastic frontier analysis approach and panel regression approach, which are included in this study. The results show Indonesia has a higher mean operating income score than Singapore, South Africa, Pakistan, and Malaysia, of a million USD. A similar pattern is also shown in the liabilities variable. In the current and fixed assets, the result remains the same, except for Malaysia and Pakistan; in these two variables, Malaysia's current and fixed assets describe higher scores than Pakistan's. The highest score for equity is Indonesia, followed by South Africa, Singapore, Malaysia, and Pakistan, with their respective values of 763,488,884; 364,656,508; 351,824,566; 123,357,947; and 53,626,094 in USD. In short, Table 2 reflects that Indonesia, Singapore, and South Africa consistently maintain their input and output variables in the halal food and beverage sector. On the other hand, Malaysia and Pakistan remain the same, with the lowest input and output variables.

Panel regression statistics descriptive information reveals that the highest average firm value is Pakistan (4.80), followed by Malaysia (2.985), Indonesia (1.809), Singapore (1.489), and South Africa (1.482). This information indicates the performance of the industry on the stock market, as well as investor wealth. Secondly, the technical efficiency score, which represents the ability of the firm to allocate their resources for creating an output, shows Indonesia has the highest score (0.620), followed by Pakistan (0.594), South Africa (0.566), Malaysia (0.550), and Singapore (0.523). Otherwise, the controlling variable, firm size, indicates Singapore has the most significant result with 20.271; the enormous size of the company indicated high operating expenses that cause the efficiency score lower than the other sample [60]. Furthermore, other controlling variables show diverse patterns of rank among the countries, such as Pakistan's highest leverage which indicate the dependency for using debt as a financing tools to support firm operational, whereas the highest inflation can be a sign as increasing price of goods and services in macroeconomic scale. While Indonesia's shows the highest growth of halal food and beverage production, means the industry has massive demand.

### Stochastic frontier analysis result

One of the influential factors in increasing efficiency in Malaysia's halal food and beverage firm is the addition of current assets to operational system. When the current assets increase by 1%, the operating income will rise by 1.0073407% and give a positive domino effect toward the efficiency score. The result indicates that current assets, such as raw materials, equipment, and inventory, must be appropriately managed to improve the company's performance [10, 12, 28, 38, 39, 41, 42]. Good management of current assets can affect the liquid of cash circular flow to the companies, as a result, they can produce other resources and maximizing their profit.

To improve efficiency performance in Indonesia, Table 3 reveals that current and fixed assets are the main supporting factors that affect company operating income [12, 28, 38, 39, 41], means when current asset increase by 1%, the technical efficiency also increase by 1.3393270% and when fixed asset going up to 1%, the firm efficiency performance will go up to 1.0045894%. The result shows that in the case of Indonesia, this industry does not yet have a qualified production support machine and need to be improved by advance technology [61]. On the other hand, debt is another factor that needs to be considered because it negatively affects the Indonesia halal food and beverage companies' performance, which means when the debt increase at 1%, the technical efficiency score will decrease at -0.9328930%. Although there

**Table 2. Descriptive statistic.**

**Stochastic Frontier Analysis Model**

| Variables | Malaysia | Indonesia | Singapore | Pakistan | South Africa |
|---|---|---|---|---|---|
| **Operating Income** | | | | | |
| Mean | 31,522,281 | 158,683,681 | 117,906,484 | 38,527,348 | 78,557,719 |
| S. Dev | 51,495,289 | 263,487,145 | 118,886,052 | 49,278,814 | 124,582,965 |
| Max | 226,031,657 | 1,227,040,173 | 532,524,000 | 213,772,463 | 691,230,000 |
| Min | 2,345,935 | 3,207,432 | 3,139,608 | 1,905,813 | 264,903 |
| **Current Assets** | | | | | |
| Mean | 123,683,944 | 534,840,833 | 462,942,221 | 81,978,425 | 267,984,721 |
| S. Dev | 129,856,558 | 813,159,936 | 476,417,707 | 59,059,683 | 433,528,294 |
| Max | 551,029,350 | 3,797,279,495 | 1,856,423,000 | 236,299,952 | 2,534,227,000 |
| Min | 24,226,147 | 19,717,264 | 12,550,888 | 18,048,378 | 2,704,214 |
| **Fixed Assets** | | | | | |
| Mean | 103,167,123 | 947,957,378 | 940,819,265 | 88,076,953 | 368,085,220 |
| S. Dev | 130,339,966 | 2,012,000,637 | 911,018,908 | 71,572,732 | 690,522,787 |
| Max | 445,778,174 | 8,842,129,537 | 2,823,814,447 | 300,826,793 | 4,303,957,000 |
| Min | 15,892,408 | 21,275,240 | 23,337,267 | 18,905,768 | 3,750,904 |
| **Liability** | | | | | |
| Mean | 103,493,119 | 719,309,327 | 589,590,346 | 116,208,562 | 271,413,433 |
| S. Dev | 152,253,179 | 1,418,150,837 | 591,007,197 | 130,711,228 | 500,184,129 |
| Max | 575,223,649 | 6,498,286,593 | 2,155,761,000 | 486,370,927 | 3,168,724,000 |
| Min | 4,684,466 | 13,280,054 | 16,564,644 | 10,586,088 | 5,071,853 |
| **Equity** | | | | | |
| Mean | 123,357,947 | 763,488,884 | 351,824,566 | 53,626,094 | 364,656,508 |
| S. Dev | 134,690,821 | 1,383,356,631 | 542,656,384 | 31,194,757 | 633,213,372 |
| Max | 673,058,737 | 6,071,349,247 | 1,779,782,315 | 142,918,983 | 3,669,460,000 |
| Min | 25,482,421 | 22,702,228 | -213,166,000 | 1,643,263 | 1,497,575 |

**Panel Regression Model**

| Variables | Malaysia | Indonesia | Singapore | Pakistan | South Africa |
|---|---|---|---|---|---|
| **Tobins'Q** | | | | | |
| Mean | 2.985 | 1.809 | 1.489 | 4.380 | 1.482 |
| S. Dev | 4.048 | 0.999 | 0.958 | 3.228 | 1.216 |
| Max | 24.583 | 3.885 | 3.983 | 12.605 | 5.544 |
| Min | 0.720 | 0.704 | 0.659 | 1.327 | 0.415 |
| **Technical Efficiency Score** | | | | | |
| Mean | 0.550 | 0.620 | 0.523 | 0.594 | 0.566 |
| S. Dev | 0.236 | 0.228 | 0.268 | 0.171 | 0.215 |
| Max | 0.943 | 1.000 | 1.000 | 0.853 | 0.904 |
| Min | 0.064 | 0.154 | 0.034 | 0.131 | 0.044 |
| **Firm Size** | | | | | |
| Mean | 18.886 | 19.860 | 20.271 | 19.093 | 19.032 |
| S. Dev | 1.216 | 1.365 | 1.122 | 0.771 | 1.700 |
| Max | 21.040 | 22.664 | 22.257 | 20.825 | 21.582 |
| Min | 13.753 | 17.681 | 17.838 | 17.795 | 16.241 |
| **Leverage** | | | | | |
| Mean | 0.340 | 0.417 | 0.443 | 0.585 | 0.457 |
| S. Dev | 0.214 | 0.152 | 0.107 | 0.254 | 0.158 |
| Max | 0.805 | 0.707 | 0.628 | 0.956 | 0.785 |

*(Continued)*

**Table 2.** (Continued)

| | | | | | |
|---|---|---|---|---|---|
| Min | 0.076 | 0.141 | 0.289 | 0.132 | 0.198 |
| **Growth** | | | | | |
| Mean | 0.039 | 0.135 | 0.077 | 0.024 | 0.042 |
| S. Dev | 0.190 | 0.252 | 0.134 | 0.149 | 0.168 |
| Max | 0.864 | 1.074 | 0.563 | 0.389 | 0.574 |
| Min | -0.999 | -0.205 | -0.095 | -0.258 | -0.288 |
| **Inflation** | | | | | |
| Mean | 1.351 | 2.704 | 0.741 | 7.796 | 4.329 |
| S. Dev | 1.714 | 0.843 | 0.841 | 2.706 | 0.663 |
| Max | 3.871 | 3.809 | 2.305 | 10.578 | 5.184 |
| Min | -1.139 | 1.560 | -0.182 | 4.085 | 3.210 |

Source: Data processed

is already a fairly high threshold regulation related to the use of debt in Islamic stocks in Indonesia, which is minimum 45% compare to total asset, it still does not have a good impact on the company's internals.

In Singapore, Table 3 shows that one of the variables that can positively influence an increase in halal food and beverage companies' performance is capital participation. The capital derived from the company's profits and investors can affect the company's performance by 0.02959797%. Means, equity is a crucial point to help companies be sustainable [10, 28, 38, 40, 42]. However, the other variable such as current asset, fixed asset, and liability indicates positive insignificant for affecting efficiency score.

Based on Table 3, to improve halal food and beverage performance in Pakistan, one of the variables that has a significant positive effect is current assets. When the current asset grows by 1%, the operating income and efficiency score will go up by 2.583678%. These results are in

**Table 3. Stochastic frontier analysis result.**

| Variables | Malaysia | Indonesia | Singapore | Pakistan | South Africa |
|---|---|---|---|---|---|
| **β0** | | | | | |
| Coefficient | 0.2806993 | -2.6069379 | 0.83613076 | 6.3910006 | -6.756681 |
| t-ratio | 0.2582096 | -2.8958532 | 0.8450488 | 1.9526283 | -6.77994 |
| **Current Assets** | | | | | |
| Coefficient | 1.0073407 | 1.339327 | 0.22243388 | 2.5383678 | 2.0786926 |
| t-ratio | 2.7316908* | 3.1055886* | 1.3227044 | 7.0174520* | 2.4004299* |
| **Fix Assets** | | | | | |
| Coefficient | 0.359721 | 1.0045894 | 0.1228251 | -0.4584897 | 0.0914668 |
| t-ratio | 1.5052318 | 2.9223852* | 0.7040556 | -1.2292113 | 0.1059019 |
| **Liability** | | | | | |
| Coefficient | 0.2167884 | -0.932893 | 0.550514 | -0.5912594 | 0.1998825 |
| t-ratio | 0.8723148 | -2.8494950* | 1.896566 | -1.9144204 | 0.3619396 |
| **Equity** | | | | | |
| Coefficient | -0.6340798 | -0.3320828 | 0.02959797 | -0.8922905 | -1.0449715 |
| t-ratio | -1.8581452 | -0.7612119 | 3.0533874* | -3.8356016* | -1.7523601 |

*significant at 5%

Source: data processed

line with. This means that Pakistan's halal food and beverage processing industry shows a good life cycle product [10, 12, 28, 38, 39, 41, 42]. The capital variable, on the other hand, shows the opposite result: that, on average, halal food and beverage companies in Pakistan are too dependent on external funding, which can decrease the efficiency score by -0.8922905% when the equity increases by 1%. If left unchecked, this can have an impact on debt that cannot be paid [53].

In South Africa, Table 3 shows that using current assets has a significant role in improving the company's operational performance, meaning that there is the ability to manage a good cash conversion cycle in this sector [10, 12, 28, 38, 39, 41, 42]. When the current assets increase by 1%, it will increase by 2.0786926% of operating income; as a result, the efficiency score will also perform better. On the other hand, equity indicates a contrary result, if the company raises their equity by 1% it, will decrease the operating income (-1.0449715) as well as efficiency score. Although the equity showed insignificant effect, the company should look at their budget line carefully to minimize the bad performance.

## Panel regression result

Table 4, shows that REM is the best model for Malaysia, which is due to the probability score of the Hausman test, which indicates 1.00 > 0.05, and means REM is the fit approach [58]. In the case of Malaysia, the results show that the technical efficiency of halal food and beverage companies had a significant effect on investor perceptions simultaneously (0.000720) and partially (0.0246). Increase in the efficiency score of 1 unit can increase investors' intention to invest in the halal food and beverage industry by 3.2 units. Furthermore, the model also indicates that the halal industry in Malaysia has a strong appeal to investors, and this is in line with the vast share of the halal market in Malaysia, whereas the high awareness of Malaysian society to consume halal food and beverage and not only for muslim community but also the other religions [54] and simultaneously proves a relationship to the signal theory, that can show that information circulating on the Islamic stock exchange in Malaysia contains information relevant to investors [21, 22].

**Table 4. Malaysia panel regression result.**

| Dependent Variable: Y_TQ | | | |
|---|---|---|---|
| Method: Panel EGLS (Cross-section random effects) | | | |
| Sample: 2017 2021 | | | |
| Periods included: 5 | | | |
| Cross-sections included: 17 | | | |
| Total panel (balanced) observations: 85 | | | |
| Variable | Coefficient | t-Statistic | Prob. |
| $\beta_0$ | -0.981138 | -0.147806 | 0.8829 |
| *Technical Efficiency* | 3.206398 | 2.290703 | 0.0246 |
| *Firm Size* | -0.058052 | -0.158641 | 0.8744 |
| *Leverage* | 10.21923 | 4.687245 | 0.0000 |
| Growth | -0.080369 | -0.052057 | 0.9586 |
| *Inflation* | -0.125324 | -0.930424 | 0.3550 |
| R-squared | 0.232316 | | |
| Prob[F-statistic] | 0.000720 | | |

Source: data processed

**Table 5. Indonesia panel regression result.**

| Dependent Variable: Y_TQ | | | |
|---|---|---|---|
| Method: Panel EGLS (Cross-section random effects) | | | |
| Sample: 2017 2021 | | | |
| Periods included: 5 | | | |
| Cross-sections included: 12 | | | |
| Total panel (balanced) observations: 60 | | | |
| Variable | Coefficient | t-Statistic | Prob. |
| $\beta_0$ | -3.028196 | -0.813187 | 0.4197 |
| *Technical Efficiency* | 0.929285 | 2.681620 | 0.0097 |
| *Firm Size* | 0.268421 | 1.441066 | 0.1553 |
| *Leverage* | -3.788711 | -5.421756 | 0.0000 |
| *Growth* | -0.151444 | -0.596343 | 0.5534 |
| *Inflation* | 0.195911 | 3.006551 | 0.0040 |
| R-squared | 0.408020 | | |
| Prob[F-statistic] | 0.000023 | | |

Source: data processed

The results of the panel data analysis for Indonesia show in Table 5, REM is the best approach. This can be seen from the Hausman Probability score (1.0000>0.05) [58]. Furthermore, the efficiency score has a simultaneous (0.000023) and partial effect (0.0097) on investors' assessments. Means that the halal food and beverage sector in Indonesia is one of the industries that is quite influential in the capital market [15, 17, 24, 52, 53]. This supports the fact that the market share of Islamic investors in Indonesia reaches 81% [2]. In addition, the results of this study support the signal theory, which states that information on the market can be one of the considerations for investors [21, 22]. In addition, managing the efficiency score can be beneficial for company and investor.

The REM model is indicated as a fit model for the panel regression model in Singapore because the probability of the Hausman test is 1.0000>0.05 [58]. The efficiency scores' effect on firm valuations shows insignificant result, which can be seen from probability values 0.217737 in Table 6. This indicates that Singapore's halal food and beverage industry is not one of the industries that has received special attention from investors. The insignificant signal of halal food and beverage companies efficiency to firm performance in stock market shows that the existence of halal stock still lack of attention from investors in Singapore due to low understanding about halal investment is evidence of this [62]. However, maintaining technical efficiency still has significant positive impact partially for the company; so if the company raises the efficiency score by 1%, firm value will increase by 0.659939%.

The best approach for a panel regression model in Pakistan is REM, due to the Hausman test indicating a 0.56 probability score, which is higher than 0.05 [58]. The results of the regression of panel data indicate that simultaneously and partially, the level of technical efficiency does not influence the value of the enterprise in the case of Pakistan, which can be seen in Table 7. This shows that investment in the halal sector in Pakistan has not grown rapidly. This result is in line with research by [55], which states that the halal industry's market share and Islamic investment are far from optimal.

CEM is the fit model for the South African regression model, which can be seen from the Lagrange Multiplier test, Breusch-Pagan, which shows 0.828>0.05 of probability [58]. The

**Table 6. Singapore panel regression result.**

| Dependent Variable: Y_TQ | | | |
|---|---|---|---|
| Method: Panel EGLS (Cross-section random effects) | | | |
| Sample: 2017 2021 | | | |
| Periods included: 5 | | | |
| Cross-sections included: 8 | | | |
| Total panel (balanced) observations: 40 | | | |
| Variable | Coefficient | t-Statistic | Prob. |
| $\beta_0$ | 6.759654 | 4.646954 | 0.0000 |
| *Technical Efficiency* | 0.659939 | 3.620300 | 0.0009 |
| *Firm Size* | -0.253939 | -3.627666 | 0.0009 |
| *Leverage* | -0.956616 | -1.852858 | 0.0726 |
| *Growth* | 0.054649 | 0.195505 | 0.8462 |
| *Inflation* | -0.066247 | -1.755576 | 0.0882 |
| R-squared | 0.180094 | | |
| Prob[F-statistic] | 0.217737 | | |

Source: data processed

influence between the technical efficiency score to firm value in Table 8 shows insignificant relationship. This indicates that the halal food and beverage industry performance is not the primary determinant stock option for investors in South Africa. This supports the research of [56], which describes the behavior intentions of investing related to the halal industry, which is still low in South Africa. In addition, [63] explained the low literacy and focus of the public regarding Islamic stocks. However, maintaining company efficiency still has a partial effect on increasing the firm value, whereas if the technical efficiency were to go up to 1%, the Tobin's Q ratio, which represent firm value and investor wealth, would increase by 2.220265%.

**Table 7. Pakistan panel regression result.**

| Dependent Variable: Y_TQ | | | |
|---|---|---|---|
| Method: Panel EGLS (Cross-section random effects) | | | |
| Sample: 2017 2021 | | | |
| Periods included: 5 | | | |
| Cross-sections included: 7 | | | |
| Total panel (balanced) observations: 35 | | | |
| Variable | Coefficient | t-Statistic | Prob. |
| $\beta_0$ | -32.60425 | -1.209757 | 0.2361 |
| *Technical Efficiency* | 1.546922 | 1.282786 | 0.2097 |
| *Firm Size* | 1.865394 | 1.309212 | 0.2007 |
| *Leverage* | 2.093301 | 0.736303 | 0.4675 |
| *Growth* | 1.211664 | 0.770387 | 0.4473 |
| *Inflation* | -0.103180 | -1.395805 | 0.1734 |
| R-squared | 0.258962 | | |
| Prob[F-statistic] | 0.104329 | | |

Source: data processed

**Table 8. South Africa panel regression result.**

| Dependent Variable: Y_TQ | | | |
|---|---|---|---|
| Method: Panel Least Squares | | | |
| Sample: 2017 2021 | | | |
| Periods included: 5 | | | |
| Cross-sections included: 8 | | | |
| Total panel (balanced) observations: 40 | | | |
| **Variable** | **Coefficient** | **t-Statistic** | **Prob.** |
| $\beta_0$ | -0.800444 | -0.293501 | 0.7709 |
| *Technical Efficiency* | 2.220265 | 2.367245 | 0.0238 |
| *Firm Size* | -0.058621 | -0.514322 | 0.6104 |
| *Leverage* | 1.265481 | 1.027958 | 0.3112 |
| *Growth* | -1.254290 | -1.001736 | 0.3235 |
| *Inflation* | 0.373466 | 1.220741 | 0.2306 |
| R-squared | 0.194566 | | |
| Prob[F-statistic] | 0.175379 | | |

Source: data processed

## Discussion

**Technical efficiency.** Looking at the detail from Table 9, the highest efficiency score from 2019 to 2021 is in Indonesia. Although the score generally declined in halal food and beverages companies in Indonesia, it was not significant. The result supports [18], which states that the consumption sector, especially basic needs, has good resistance to economic shocks, and especially had, in 2020 and 2021 when an economic outbreak existed. In terms of regulations, the Indonesian government's policy related to the halal product guarantee and the existence of the Indonesian Sharia Economic Master Plan, helped accelerate the growth of the halal industry. Indonesia has a well-developed ecosystem, especially in regulating halal products and Islamic Finance [64]. This is due to Indonesia having the highest Muslim population in the world and occupying second place as an influential country in the halal food and beverage sectors. As a result, the awareness and understanding of society about halal consumption needs in this country are high and have become the most consideration, due to the Indonesian know that shariah law only allowed them to consume something that do not consist harmful for the consumers, because of that the demand for halal food and beverages is high [43].

Next, the results of technical efficiency in Pakistan show fluctuations, starting in 2017 and moving to 2019, which experienced a decline, but in 2020 there was an insignificant increase.

**Table 9. Rank of technical efficiency score.**

| Country | 2017 | 2018 | 2019 | 2020 | 2021 | Mean | Rank |
|---|---|---|---|---|---|---|---|
| **Malaysia** | 0.552 | 0.553 | 0.594 | 0.536 | 0.513 | 0.55 | 4 |
| **Indonesia** | 0.628 | 0.55 | 0.654 | 0.638 | 0.632 | 0.62 | 1 |
| **Singapore** | 0.492 | 0.558 | 0.434 | 0.561 | 0.571 | 0.523 | 5 |
| **Pakistan** | 0.64 | 0.621 | 0.523 | 0.591 | 0.596 | 0.594 | 2 |
| **South Africa** | 0.523 | 0.542 | 0.612 | 0.599 | 0.554 | 0.566 | 3 |
| **Mean** | 0.567 | 0.5648 | 0.5634 | 0.585 | 0.5732 | 0.5706 | |

Source: data processed

However, halal food and beverage companies performed better during the economic crisis. This means that the halal industry in Pakistan is not only defensive but can also develop well amid economic instability [65]. In 2017, Pakistan's halal food and beverage industry efficiency was the highest among the four other countries considered in this paper. The main reason is that in March 2016, the Pakistani government established the Pakistan Halal Authority, which was tasked with promoting and making legal standardizations related to the halal food industry [44]. This policy has resulted in high public demand for halal products due to awareness and perception, which have increased significantly. As a result, in 2018–2019, food and beverage companies strive to meet the halal standards set by the government and optimize their performance to meet market needs [45, 66]. In short, the input and output cycle in the Pakistan halal food and beverages industry is fast, which means a firm can control its cash conversion cycle significantly, which can be seen from Table 3, in that current assets become a crucial factor in boosting technical efficiency score [10, 12, 28, 38, 39, 41, 42].

Based on Table 9, in South Africa efficiency scores over the past five years tended to fluctuate. The technical efficiency score showed a downward trend during the economic crisis from 2020 to 2021. These results indicate that the position of halal food and beverage companies in South Africa is not included in the defensive industry and not settled enough due to the firm performance in capital market that can distracted by external condition such as covid-19 economic crises, which is in line with the research of [67, 68] in different countries. The supply chain management for the halal industry is pricy. Therefore, the companies focusing on producing halal products need more equity for increasing their profitability. When the pandemic came, resources were limited due to government regulations or the price of raw materials. Furthermore, research from [56], who stated that the norms and behavior of South African society re purchasing halal products, still needs to be higher. Although the current asset becomes the foundation for increasing halal food and beverages efficiency, if the company can manage their conversion cycle from output to income, they have an excellent allocation of resources.

Furthermore, the technical efficiency score for The Malaysian halal food and beverages companies showed a relatively stable at an average of 0.55%. Even during the economic shocks from 2020 to 2021, the decline in the average efficiency score was not very significant This result is in line with [18], who found that companies in the consumption sector have a good level of defense when there is economic instability. However, the average efficiency score, which is 0.550, indicates that, on average, halal food and beverage companies in Malaysia only utilize the resources they have to produce an output of 55%, of which the remaining 45% indicate indications of inefficiency. In Malaysia, the TE score before the pandemic, increased gradually because of the highest demand for halal products, which can be seen from the significance of current assets. The Muslim society's awareness of consuming halal product is massive. Therefore, the government takes serious action to facilitate the industry's certification, by supporting the halal logo. Unfortunately, during covid-19, the TE score declined continuously, which is in line with [69] who found that the manufacturing industry sector, especially for consumer goods, is fighting difficulties due to restricted economic activity. Furthermore, the increase in lower class society, which has income under the minimum wages, might be affects lower purchasing power.

Singapore's halal food and beverage companies showed fluctuating technical efficiency performance in the first three years from 2017 to 2019. Meanwhile, during the Covid-19 economic recession halal food and beverages companies in Singapore showed a positive trend in the last two years, 2020 to 2021. This result indicates that the economic crisis is a good thing for industries engaged in the consumer goods sector, especially in basic needs products [65]. The country maximizes the use of technology to distribute its product [output], especially the basic needs such as food and beverages, as well as adds equity to the production process by buying

new machine that support the production process [65]. Therefore during the pandemic, halal food and beverage companies also tried to innovate their system to fulfill the market demand, not only domestically but also globally. However, the average efficiency of halal food and beverage companies in Singapore is the lowest because, in terms of capital in food and beverages, halal companies in Singapore are still quite strong compared to other countries, even though in terms of equity, it has a significant positive influence to improve technical efficiency. Given that TE is a function of production and capital is one of the old factors that cannot be ignored, companies in Singapore need to raise capital to increase the value of their efficiency. Some previous studies have shown that the greater the capital, the more efficient, because large capital can help with daily production operations [10, 28, 38, 40, 42].

In addition, Malaysia, Indonesia, Pakistan, and South Africa show that current assets are a crucial factor in improving a company's technical efficiency; this is due to the good quality of asset turnover, which is the process of creating the profit by using asset that owned by firm. The cash conversion cycle (CCC) theory states that ceteris paribus, efficient working capital management, which is a short CCC, will increase the company's liquidity, profitability and value. In contrast, inefficient working capital management and a long CCC, will lead to lower profitability and company value [70]. The study of the CCC is significant for the agricultural and food industries. Unlike other businesses, the food and beverage industries have unique risk factors. In addition to the general risks (access to growth capital and competition), specific risks associated with nature and macroeconomic factors (biological and weather-related risks, commodity price volatility, infrastructure in rural areas and government policies) can significantly affect the profitability of this sector [71]. A good company can generate sales by using its total assets so that it has a positive impact on revenue.

On the other hand, during the Covid-19 pandemic, which started in 2020, online selling became one of the effective methods to boost the CCC. The e-commerce trend for food and beverage retail companies increased significantly in Singapore and Pakistan; this implicated in the firm's productivity [72], which can be seen in the rising TE in 2020 and 2021. Other cases in Indonesia, Malaysia, and South Africa show a downward trend in TE during the pandemic, even though it was insignificant. It means online selling exists and helps the company to operate and create products as it fulfills the basic needs of society [65]. Furthermore, fixed assets can support generating inventory and fixing asset turnover, which positively impacts the company because it uses its assets to continue generating income.

**The effect of technical efficiency toward Tobin's Q performance.** One of the tools that can be represent the firm value and investor perspective is Tobin's Q. Knowing Tobin's Q ratio can identify the company performance from both external and internal perspectives; that is why this variable is used as a dependent variable. At the same time, technical efficiency describes the company's ability to manage its resources to generate revenue. Table 9 provides the average efficiency score for each country, which is between 0.52 and 0.62. So, it means the efficiency is quite good but still needs managerial improvement. Based on Table 10, the company's ability to manage its assets positively impacts Tobin's Q performance. The company's efficiency can increase the value of halal food and beverage companies in the stock market, especially in Malaysia, Indonesia, Singapore, and South Africa. This study's results align with the signaling theory, which states that the better the company's performance, the more it can increase market value (H1 hypothesis accepted). In other words, investors have optimistic thoughts on good efficiency stocks. The results of this study support the statement that increasing efficiency means that the company is increasingly benefitting its investors, which is illustrated in the increase in the value of shares in the market [15, 17, 24, 52, 53]. Furthermore, the result also indicates that the sharia capital market for halal food and beverage stocks has symmetric information between internal parties (company) and external parties (investor).

**Table 10. Panel regression result.**

| Name of Country | Name of Variable | Result |
|---|---|---|
| Malaysia | Technical Efficiency | Positive Significant |
| | Firm Size | Negative Insignificant |
| | Leverage | Positive Significant |
| | Growth | Negative Insignificant |
| | Inflation | Negative Insignificant |
| Indonesia | Technical Efficiency | Positive Significant |
| | Firm Size | Positive Insignificant |
| | Leverage | Negative Significant |
| | Growth | Negative Insignificant |
| | Inflation | Positive Significant |
| Singapore | Technical Efficiency | Positive Significant |
| | Firm Size | Negative Significant |
| | Leverage | Negative Insignificant |
| | Growth | Positive Insignificant |
| | Inflation | Negative Insignificant |
| Pakistan | Technical Efficiency | Positive Insignificant |
| | Firm Size | Positive Insignificant |
| | Leverage | Positive Insignificant |
| | Growth | Positive Insignificant |
| | Inflation | Negative Insignificant |
| South Africa | Technical Efficiency | Positive Significant |
| | Firm Size | Negative Insignificant |
| | Leverage | Positive Insignificant |
| | Growth | Negative Insignificant |
| | Inflation | Positive Insignificant |

Source: data processed

Regarding control variables, investors from Malaysia, Pakistan, and South Africa consider that debt is a positive thing in the business world, to boost business operations as capital. Investors do not think negatively about debt, so the company's value remains positive. In the case of Malaysia, Leverage became a significant factor for halal food and beverages companies, which can be a positive sign for increasing firm value. When a company does incorporate action to increase its debt, investors think it has good prospects for the future. Therefore, investors choose to keep their shares for a long-term investment. It indicates, because of their confidence in facing higher risk from the amount of company debt, that most halal food and beverage shareholders are risk-takers.

On the other hand, leverage became a negative sign for Indonesian and Singaporean investors. The highest leverage ratio and risk should be certified by investors. So, the behavior of shareholders for halal food and beverage companies in Indonesia and Singapore is mostly risk-averse. Indonesian companies rely on external funding sources to cover their funding needs [73]. Based on funding sources, according to Pecking Order Theory, investors tend to pay more attention to leverage than equities. Investors reacted negatively to an increase in equity size and reacted positively to a reduction in leverage [74]. The issuance of equity indicates that the company is in severe financial difficulties and overloaded [75]. Suppose the debt level is too high and the company does not use it to support operational activities, then the company will continue to pay the debt burden, so it cannot optimize quality profits [76].

In the case of Singapore, firm size negatively affects investor perspective. The huge size of a company can be a bad indicator for the investor. A big company can easily access external debt because they have capable collateral. Unfortunately, having a debt can sometimes cause default risk. Therefore, Singapore investors prefer to prevent the risk. Furthermore, the macroeconomic condition, inflation, became a significant factor in increasing the firm value in the case of Indonesia during research period. Inflation can be an interpretation of high number of transactions which occur in the market. Other cases in Pakistan and South Africa indicate that control variables are insignificant. Investors focus on the performance of production; they do not analyze the other financial ratio such as leverage, which can be cause default risk, even under macroeconomic conditions.

## Conclusion and policy implication

This study aims to analyze the impact of technical efficiency scores on a company's value, which is the investors' perception when making investment decisions, by using a sample of five countries that are influential countries in the halal food and beverage industry. The results show a difference in the pattern of efficiency scores and their effect on company value. The average efficiency score is in the range from 0.5 to 0.6. Indonesia is the country with the highest efficiency level and is followed by Pakistan, South Africa, Malaysia, and Singapore. During the economic crisis from 2020 to 2021, the technical efficiency scores of halal food and beverage companies in Malaysia, Indonesia, and South Africa experienced a downward trend, even though they were not too significant, so this was still classified as defensive against economic recession conditions. Meanwhile, Singapore and Pakistan showed the opposite trend, where there was an increase in company performance during the economic crisis; this proves that the halal food and beverage sector is a crucial industry in supporting the needs of people.

The results show that Malaysia and Indonesia, the two countries with the largest Muslim populations in the world that technical efficiency scores significantly positively impact company value simultaneously and partially. The results of this research indicate the rational behavior of Muslim investors and the symmetrical information on the stock exchanges in the two countries. Meanwhile, in Singapore, Pakistan, and South Africa, although they showed partial positive results regarding the effect of technical efficiency on investor perceptions, they did not simultaneously have a significant effect. This result indicates that the halal industry in the three countries has a not too wide market share, and suffers from insufficient attention of investors to Islamic stocks. The companies need to maintain their efficiency, even though they have received the halal label, because it has an impact on the investor perspective. Furthermore, given the potential and impact of the halal industry, it would be very good if countries that already have a halal industry consider having a capital market for these companies. It can be beneficial for companies to get external financing, and in a domino effect, can boost the economic growth of country.

This research can provide benefits for several parties. Firstly, practitioners such as companies will be able to improve their efficiency scores by adding other sources like machine and vehicle which can increase their productivity and distribution process, especially current assets, which indicates the ability of firm to manage their cash conversion cycle that all five countries have a high positive significant result. Furthermore, the TE score can be a benchmark for similar industries to improve their performance. From the investor side, the efficiency score can be used to make investment decisions, especially in Indonesia and Malaysia. Secondly, it can be helpful to compare performance between countries that can affect company and government policies related to their countries' comparative advantages. The result can give reference for the government to make the rules which can increase halal food and

beverage efficiency. Thirdly, investors can use the TE score as preference before choosing the sharia stocks, because it can be beneficial for their stock return. Lastly, for theoretical implications, this paper strongly supports the signaling theory and rational investor behavior which can be an input for other researchers, especially in the field of halal industry and Islamic capital market.

Although the sample consists of five influential halal food and beverage countries, the model can be used for other countries to calculate technical production efficiency as well as to look for the effect on investor perspective as long as the country has an Islamic capital market index, which represents sharia compliance both from the company product and financial managerial. Unfortunately, this research has limitations related to research sample, which only took five out of 11 halal influential food and beverage countries due to six of the countries still needing an Islamic capital market index. Furthermore, the existence of economic recession conditions, resulted in the data having a large gap. Therefore, further research is expected to add other variables, especially religion-related ones, such as the number of Muslim investors and macroeconomics, or add to the number of countries as research samples. However, research topics related to the halal industry and Islamic stocks are still rarely discussed simultaneously, using samples from various countries.

## Supporting information

**S1 Data.**
(XLSX)

**S2 Data.**
(XLSX)

**S1 File.**
(DOCX)

**S2 File.**
(DOCX)

## Author Contributions

**Conceptualization:** Sylva Alif Rusmita, Siti Zulaikha, Nur Syazwani Mazlan.

**Data curation:** Nur Syazwani Mazlan.

**Formal analysis:** Sylva Alif Rusmita, Nur Syazwani Mazlan.

**Funding acquisition:** Sylva Alif Rusmita, Siti Zulaikha.

**Investigation:** Sylva Alif Rusmita.

**Methodology:** Sylva Alif Rusmita.

**Resources:** Eko Fajar Cahyono.

**Software:** Indria Ramadhani.

**Supervision:** Siti Zulaikha, Nur Syazwani Mazlan, Nuradli Ridzwan Shah Bin Mohd Dali, Eko Fajar Cahyono.

**Validation:** Nur Syazwani Mazlan, Nuradli Ridzwan Shah Bin Mohd Dali.

**Writing – original draft:** Sylva Alif Rusmita, Eko Fajar Cahyono, Indria Ramadhani.

**Writing – review & editing:** Sylva Alif Rusmita, Nuradli Ridzwan Shah Bin Mohd Dali, Indria Ramadhani.

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
