## [Decision Letter · Decision Letter 0]

12 Dec 2022

PONE-D-22-31308EFFICIENCY LEVEL OF FOOD AND BEVERAGE COMPANIES AND MARKET VALUE IN SELECTED COUNTRIESPLOS ONE

Dear Dr. Rusmita,

Thank you for submitting your manuscript to PLOS ONE. After careful consideration, we feel that it has merit but does not fully meet PLOS ONE’s publication criteria as it currently stands. Therefore, we invite you to submit a revised version of the manuscript that addresses the points raised during the review process.

We look forward to receiving your revised manuscript.

Kind regards,

Asst. Prof. Dr. Nemer Badwan

PhD in Economics and Finance

Assistant Professor of Economics and Finance

Academic Editor

PLOS ONE

Journal Requirements:

2. PLOS ONE does not copy edit accepted manuscripts (https://journals.plos.org/plosone/s/criteria-for-publication#loc-5). To that effect, please ensure that your submission is free of typos and grammatical errors.

3. Please ensure that you include a title page within your main document. We do appreciate that you have a title page document uploaded as a separate file, however, as per our author guidelines (http://journals.plos.org/plosone/s/submission-guidelines#loc-title-page) we do require this to be part of the manuscript file itself and not uploaded separately.

4. We note you have included a table to which you do not refer in the text of your manuscript. Please ensure that you refer to Tables 2, 3, 5, 6, 7, 8, 9, 11, 12, 14, 15, 16, 17, 18, and S1 in your text; if accepted, production will need this reference to link the reader to the Table.

Reviewers' comments:

Reviewer's Responses to Questions

**Comments to the Author**

1. Is the manuscript technically sound, and do the data support the conclusions?

Reviewer #1: No

Reviewer #2: Yes

2. Has the statistical analysis been performed appropriately and rigorously? 

Reviewer #1: No

Reviewer #2: Yes

3. Have the authors made all data underlying the findings in their manuscript fully available?

Reviewer #1: Yes

Reviewer #2: Yes

4. Is the manuscript presented in an intelligible fashion and written in standard English?

Reviewer #1: Yes

Reviewer #2: No

5. Review Comments to the Author

Reviewer #1: Review report PONE-D-22-31308

Efficiency level of food and beverage companies and market value in selected countries

Topic

The author should highlight Islamic (Halal) food and beverage companies in the topic.

Introduction

Row 72, halal food and beverage companies in the 11 best countries. Best countries in terms of?

Literature Review

It is inappropriate to include current assets, fixed assets, liabilities, equity, operating income, firm value as the sub-sections under the Literature Review section. Since they are the variables of this study, it is more appropriate to discuss them under the Methodology section.

Previous research

The authors should include the literature on the food and beverage industry.

Hypothesis

H1 is not a hypothesis. It is just an interpretation of the efficiency score. Please remove it.

Usually, each hypothesis is developed from a theory. H2 is developed from which theory?

Method

Row 241, the paper mentions the top 11 halal food countries. Row 244, it mentions the top 5 countries. So it is 11 or 5? Inconsistent data presentation. These 51 firms are listed companies? Please provide a list of company names as an appendix. The authors should mention which software they use to compute the results for step 1 and step 2.

Result

Since the authors report the result for each country, there should be a list of sample size for each country under the method section.

Row 330, showed a relatively stable result of between 0.513 and 0.55. Is this information obtained from Table 1? Please elaborate it.

The authors show so many results in different tables, but the text does not elaborate on the content well. Please improve it.

Table 3, 6, 7, 9, 10, 12, 13 and 15. The authors should not write the coefficient and independent symbol together. That is not the appropriate way to present the panel regression model.

Row 503, Table 18, it is inappropriate to report the p-value under the result column. Which text is presenting Table 18?

Conclusion

Since the authors do not present the result and discussion sections well, it is difficult to comment on the conclusion section. Therefore, the authors need to put more effort to improve the result and discussion sections.

Reviewer #2: The authors try to examine the effect of Technical Efficiency on firm value for the top halal food and beverage firms in 5 countries. They show that improvements in technical efficiency scores positively impact firm values. This study seems important as it studies an underserved topic of halal food and beverages that have not been studied in detail, yet.

Here are my suggestions to the authors.

1. The study needs some reformatting in the structure. There are numerous guides on how to format an academic paper. I suggest the authors read Bellemare (2020) and reformat the paper as much as possible.

2. The study needs a revamp of the English language formatting. The authors can benefit from sending it to a professional editor to reformat the sentences and paragraphs. While the content of the study is important, it seems like it is written in a very simple language. While there are numerous examples, I will just focus on three simple ones. In the first paragraph of the introduction the authors write “The market share of Muslims in the world as of 2019 was 1.9 billion,”. I believe they meant the number of Muslims, not market share. Then in the second page the authors write “The paper seeks to fill the gap in some of these studies”. I believe the authors mean “fill the gap of the literature”. In page the authors write “The results of this study can be helpful for all parties.” Please explain what these parties are in this sentence.

3. Please add a short paragraph at the end of the introduction telling the readers what to expect in the coming section, such as “In section two, we review the literature, while we discuss the methodology in the following section….”

4. Please merge sections Literature Review and Previous Research.

5. The authors need to update the literature review to include the following studies. Henderson (2016) discusses the impact that certification of halal food has on tourism, while Secinaro and Calandra (2020) conducts a detailed review of literature on halal food, which will help the authors in revamping their literature section. In addition, Balli et al (forthcoming) quantifies the extreme connectedness between agricultural commodity prices with food and beverage sector stock market returns.

6. There is no need to explain β_0 in page 10.

7. The sections Result and Discussion and Result, should be Results and Discussion, and Results.

8. Please remove tables 2, 5, 8, 11. They do not add anything to the academic discussion. They are just the data of the paper.

9. In tables 1, 4, 7, 10, please remove T-ratio. You have already reported Standard Error. No need to report them both. Alternatively, you can drop standard error column, and keep T-ratio.

10. In tables 3, 6, 9 and 12, you do not have to report all those statistics under “Effects Specification” part. Adding one of the R-squares and number of observations should be enough.

11. The authors need to think about generalizability of these results. Can these findings be used for other countries? At what conditions? Maybe they can think in this study, or future studies if not possible here, to do a panel regression of all firms, in all countries, and they can control for macro levele variables, or share of the Muslim population in the country, or maybe some variable where they measure observability of the religion.

12. Please remove Supporting Info part of the paper. It is not relevant for an article at PlosONE.

Referenes

Bellemare, M. (2020) “How to Write Applied Papers in Economics” Working paper http://marcfbellemare.com/wordpress/wp-content/uploads/2020/09/BellemareHowToPaperSeptember2020.pdf

Joan C. Henderson, (2016) Halal food, certification and halal tourism: Insights from Malaysia and Singapore, Tourism Management Perspectives, Volume 19, Pages 160-164, ISSN 2211-9736,

https://doi.org/10.1016/j.tmp.2015.12.006.

Secinaro, S. and Calandra, D. (2021), "Halal food: structured literature review and research agenda", British Food Journal, Vol. 123 No. 1, pp. 225-243. https://doi.org/10.1108/BFJ-03-2020-0234

Balli, Faruk, Billah, Mabruk, Hoxha, Indrit (forthcoming) “Extreme Connectedness of Agri-Commodities with Stock Markets and its Determinants”, Global Finance Journal, forthcoming.

6. PLOS authors have the option to publish the peer review history of their article (what does this mean?). If published, this will include your full peer review and any attached files.

Reviewer #1: No

Reviewer #2: No

---

## [Author Response · Author response to Decision Letter 0]

29 Jan 2023

Dear Editorial team and reviewers, thank you for your concern about this article. Here we send the response from the previous review and link to access the raw database as well as a result: https://doi.org/10.6084/m9.figshare.21971270 (we also put the link at the cover letter and supporting information). We re-wrote the table information; please look at the "manuscript" file.

Topic

The author should highlight Islamic (Halal) food and beverage companies in the topic.

We have revised with highlight the Halal term in title and empirical review (line 252-294).

Introduction

Row 72, halal food and beverage companies in the 11 best countries. Best countries in terms of?

We have revised by adding information on the 5 largest halal food producing countries based on the 2020 Global Islamic Economic Report (line 79-81). We have revised from 11 countries to 5 countries according to the research methods chapter.

Literature Review

It is inappropriate to include current assets, fixed assets, liabilities, equity, operating income, firm value as the sub- sections under the Literature Review section. Since they are the variables of this study, it is more appropriate to discuss them under the Methodology section.

We have revised it by adding this section to the research method section (line 342-375) (line 423-439)

Previous research

The authors should include the literature on the food and beverage industry.

We have added a review of 6 articles, namely Henderson (2016) and Secinaro & Calandra (2020), Putri et al (2020), Bhaskaran (2021), Mehraliana (2012), and Cummins (2009) We have written about it at the beginning of the empirical review chapter (line 175-238)

Hypothesis

H1 is not a hypothesis. It is just an interpretation of the efficiency score. Please remove it. Usually, each hypothesis is developed from a theory. H2 is developed from which theory?

We have revised according to the comments. We rewrite the hypothesis part and use the H2, which based on signaling theory hypothesis. We think this hypothesis is relevant because of our paper looking for what the effect of having good efficiency toward firm value (line 276-286)

Method

Row 241, the paper mentions the top 11 halal food countries. Row 244, it mentions the top 5 countries. So it is 11 or 5? Inconsistent data presentation. These 51 firms are listed companies? Please provide a list of company names as an appendix. The authors should mention which software they use to compute the results for step 1 and step 2.

We use the top 5 halal food and beverage companies which take from the global islamic economy report 2020. Then, we do purposive sampling by ignore the country which do not have sharia index from 2017 to 2021 (294-301). For the final sample, we got 5 countries and all of the companies are listed companies. For company name and stock code, we put on the appendix table. We used Frontier 4.1 to calculate the technical efficiency score and Eviews 9 for panel data regression. (We edit following your suggestion in line 310-314)

Result

Since the authors report the result for each country, there should be a list of sample size for each country under the method section.

We adding the table related to list of countries in line 304. List of sample for each countries also explain at panel regression result (table 7-11)

Row 330, showed a relatively stable result of between 0.513 and 0.55. Is this information obtained from Table 1? Please elaborate it.

We rewrite this section for each country at discussion section, technical efficiency part (line 638-718). All the data in this section come from table 12.

The authors show so many results in different tables, but the text does not elaborate on the content well. Please improve it.

We try to fix the result and discussion section. Each sections we separate into 2 points, first, explain about technical efficiency and second, about panel data related (the effect of efficiency toward firm value). For result section, we explain the data and supporting result from previous study. And for discussion, we analyze the result based on fact and theory which support our finding.

Table 3, 6, 7, 9, 10, 12, 13 and 15. The authors should not write the coefficient and independent symbol together. That is not the appropriate way to present the panel regression model.

Thanks for suggestion. We follow your instruction, please take a look at table 2-11

Row 503, Table 18, it is inappropriate to report the p-value under the result column. Which text is presenting Table 18?

We edit this part, and explain the research column by “significant” or “insignificant”. Please take a look at the new table (table 13, line 731). 

Conclusion

Since the authors do not present the result and discussion sections well, it is difficult to comment on the conclusion section. Therefore, the authors need to put more effort to improve the result and discussion sections.

We summarized the result and discussion at conclusion and explain about the implication for companies, investors, government, and future research at implication.

Reviewer #2: The authors try to examine the effect of Technical Efficiency on firm value for the top halal food and beverage firms in 5 countries. They show that improvements in technical efficiency scores positively impact firm values. This study seems important as it studies an underserved topic of halal food and beverages that have not been studied in detail, yet.

Here are my suggestions to the authors.

1. The study needs some reformatting in the structure. There are numerous guides on how to format an academic paper. I suggest the authors read Bellemare (2020) and reformat the paper as much as possible.

We have revised this comment, that is, we follow the chapter numbering and chapter naming according to the Bellemare reference, for example, we changed the grand theory chapter to the theoretical framework chapter, changed the previous study chapter title to empirical review, changed the method chapter title to data and statistical method and so on.

2. The study needs a revamp of the English language formatting. The authors can benefit from sending it to a professional editor to reformat the sentences and paragraphs. While the content of the study is important, it seems like it is written in a very simple language. While there are numerous examples, I will just focus on three simple ones. In the first paragraph of the introduction the authors write “The market share of Muslims in the world as of 2019 was 1.9 billion,”. I believe they meant the number of Muslims, not market share. Then in the second page the authors write “The paper seeks to fill the gap in some of these studies”. I believe the authors mean “fill the gap of the literature”. In page the authors write “The results of this study can be helpful for all parties.” Please explain what these parties are in this sentence.

We did the proofread and re-edit unnecessary sentences. Explanations regarding several parties have been written in the following sentences. The Parties are halal food companies, investors, government, and researcher (line 88-10).

3. Please add a short paragraph at the end of the introduction telling the readers what to expect in the coming section, such as “In section two, we review the literature, while we discuss the methodology in the following section….”

We have added a description of the scheme for the preparation of the sections of this research at the end of the introduction section. (line 117-125)

4. Please merge sections Literature Review and Previous Research.

We apologize for not being able to combine the two because we followed the advice from Reviewer to name and give chapter numbers based on Bellemare's paper (2020). We thank you for your input and comments.

5. The authors need to update the literature review to include the following studies. Henderson (2016) discusses the impact that certification of halal food has on tourism, while Secinaro and Calandra (2020) conducts a detailed review of literature on halal food, which will help the authors in revamping their literature section. In addition, Balli et al (forthcoming) quantifies the extreme connectedness between agricultural commodity prices with food and beverage sector stock market returns.

We have added a review of 2 articles, namely Henderson (2016) and Secinaro & Calandra (2020). We have written about it in line 210-229 of the empirical review chapter. We apologize that we could not find the Bali et al article anywhere and we replaced it with another relevant article (Cummins, 2009; Mehraliana, 2012; Bhaskaran, 2021; and Putri, 2020). We thank you for your input and comments.

6. There is no need to explain β_0 in page 10.

Thanks for the suggestion, we remove the β0 at line 335 and 395

7. The sections Result and Discussion and Result, should be Results and Discussion, and Results.

We separate the result and discussion section. For result, it explain about the data. And discussion, we combine between data that we got with theoretical or previous result which are in line to support the research finding.

8. Please remove tables 2, 5, 8, 11. They do not add anything to the academic discussion. They are just the data of the paper.

We follow your advise to delete this part. 

9. In tables 1, 4, 7, 10, please remove T-ratio. You have already reported Standard Error. No need to report them both. Alternatively, you can drop standard error column, and keep T-ratio.

We delete the t-ratio following your suggestion. (table 2-6.

10. In tables 3, 6, 9 and 12, you do not have to report all those statistics under “Effects Specification” part. Adding one of the R-squares and number of observations should be enough.

Thanks for the advise. We follow your instruction. (Table 7-11)

11. The authors need to think about generalizability of these results. Can these findings be used for other countries? At what conditions? Maybe they can think in this study, or future studies if not possible here, to do a panel regression of all firms, in all countries, and they can control for macro level variables, or share of the Muslim population in the country, or maybe some variable where they measure observability of the religion.

It is possible for other country to use the same model to calculate the efficiency and for panel data model, future research can add the other controlling variables such as number of muslim population or other macroeconomics variable (Consumption, GDP). We explain this in line 831-840.

12. Please remove Supporting Info part of the paper. It is not relevant for an article at PlosONE.

We follow your suggestion to delete the support information and put another table (The List of company name) following the suggestion from 1st reviewer.

---

## [Decision Letter · Decision Letter 1]

13 Feb 2023

PONE-D-22-31308R1EFFICIENCY LEVEL OF HALAL FOOD AND BEVERAGE COMPANIES AND MARKET VALUE IN SELECTED COUNTRIESPLOS ONE

Dear Dr. Sylva Alif Rusmita,

Thank you for submitting your manuscript to PLOS ONE. After careful consideration, we feel that it has merit but does not fully meet PLOS ONE’s publication criteria as it currently stands. Therefore, we invite you to submit a revised version of the manuscript that addresses the points raised during the review process.

We look forward to receiving your revised manuscript.

Kind Regards,

Asst. Prof. Dr. Nemer Badwan

PhD in Economics and Finance

Assistant Professor of Economics and Finance

Academic Editor

PLOS ONE

Reviewers' comments:

Reviewer's Responses to Questions

**Comments to the Author**

1. If the authors have adequately addressed your comments raised in a previous round of review and you feel that this manuscript is now acceptable for publication, you may indicate that here to bypass the “Comments to the Author” section, enter your conflict of interest statement in the “Confidential to Editor” section, and submit your "Accept" recommendation.

Reviewer #3: All comments have been addressed

Reviewer #4: (No Response)

2. Is the manuscript technically sound, and do the data support the conclusions?

Reviewer #3: Yes

Reviewer #4: Partly

3. Has the statistical analysis been performed appropriately and rigorously? 

Reviewer #3: Yes

Reviewer #4: No

4. Have the authors made all data underlying the findings in their manuscript fully available?

Reviewer #3: Yes

Reviewer #4: Yes

5. Is the manuscript presented in an intelligible fashion and written in standard English?

Reviewer #3: Yes

Reviewer #4: No

6. Review Comments to the Author

Reviewer #3: Dear Authors,

thank you for addressing the previous elements.

What I can suggest to fix and create a better conclusion and implications section.

Particularly, what I expect is a holistic section with some sub-paragraphs as Theoretical Implications and Practical Implications.

Then, more practical implications should be added.

All the best

The reviewer

Reviewer #4: Title: I suggest that the title of this manuscript should be “The impact of technical efficiency on the Firms’ value: the case of the Halal food and beverage industry in selected countries”.

Abstract: This section has been descriptively written. It is suggested to include the findings of the study here. In addition, policy implications should be included concisely at the end of this section.

Introduction: Data used in this section should be updated to most recent years. Organization of remaining of this study should be outlined at the end of this section.

Literature Review: This section should be divided into three parts. The first part must be based on the theoretical background. Secondly, the discussion should be developed on the estimation of technical efficiency. DEA and SFA approaches are used to estimate technical efficiency. This study employed the SFA approach to calculate the technical efficiency of halal food and beverages. Please justify the SFA approach compared to the DEA approach. Consult the latest published articles for your guidance and improve this section (https://doi.org/10.3390/en14133923;
https://doi.org/10.1007/s10668-020-01023-2;
https://doi.org/10.3389/fenrg.2022.943771;
https://doi.org/10.1007/s10668-022-02194-w). Third, include empirical literature that explores the impact of efficiency on firm value. Finally, find the research gap that this study fills.

Data and Statistical Method: (1). The title of this section should be “Methodology”. (2). It is recommended to include a table regarding “description of the data series” used in this study. (3). It is recommended to justify the significance of each variable included in the model. (4). Technical efficiency is estimated from the production function including capital, labor and other factors of the firm. This study is missing labor of the firm.

Results and Discussion: first, it is recommended to include descriptive statistics of the variables for each country used in the study in tabular form. This study is based on panel data with five cross sections and time period 2017-2021. Use the panel data SFA model to calculate the technical efficiency of each country with different time periods. It is recommended to use the above-mentioned articles for your guidance and to improve this section. Second, report findings of the SFA panel data model along with diagnostic statistics. Third, report each country's technical efficiency along with their ranking. Finally, to explore the impact of technical efficiency and other control variables on firm performance, use panel data models such as fixed effect, random effect, ARDL panel, or fully modified dynamic panel model, whichever best fits your data series. It is recommended to merge the discussion section with the "Results and Discussion" section.

Conclusion: This section should be titled “Conclusions and policy Implications”. Include policy implications based on study results. It is recommended to include “Future Research Directions” at the end of this section.

7. PLOS authors have the option to publish the peer review history of their article (what does this mean?). If published, this will include your full peer review and any attached files.

Reviewer #3: No

Reviewer #4: **Yes: **Prof. Dr. Dilawar Khan

<quillbot-extension-portal></quillbot-extension-portal>

---

## [Author Response · Author response to Decision Letter 1]

16 May 2023

Dear reviewers

Thanks for giving us precious feed back to improve this manuscript. Here our answer for each advise that you give to us, we also attached this part by file entitled "Response to reviewers"

Reviewer #3: Dear Authors,

thank you for addressing the previous elements.

What I can suggest to fix and create a better conclusion and implications section.

Particularly, what I expect is a holistic section with some sub-paragraphs as Theoretical Implications and Practical Implications. Then, more practical implications should be added. 

All the best

The reviewer

-Thanks for the suggestion, we explain this part at 3th paragraph of conclusion and policy implication section. For theoretical implication, this research strongly support the signaling theory and emphasize that investor in halal FnB consider as rational investor. Along with that, we also explain the practical implication for several parties such as firm which can be known their operating managerial, benchmark, and remain them to control their TE score, because it is impacting to firm performance in capital market. Furthermore, the implication for policy maker to make the rules like giving intensive to the industry to minimizing the inefficiency that happen in firm, consider the strongest of halal FnB compare to other countries so they can know the comparative advantages. At the same time this research also beneficial for investor to use TE score as consideration to make decision making because it can impact their wealth as well.

Reviewer #4: Title: I suggest that the title of this manuscript should be “The impact of technical efficiency on the Firms’ value: the case of the Halal food and beverage industry in selected countries”.

We re-name the title following your advice.

Abstract: This section has been descriptively written. It is suggested to include the findings of the study here. In addition, policy implications should be included concisely at the end of this section.

-Thanks for the suggestion. We edit the result by mention each country efficiency score and explain the inefficiency score as well which indicate this industry need further improvement especially form companies managerial side. For policy implication we emphasize this research can beneficially help government to support this sector such as giving subsidy to boost their performance and to identify the strength of industry by comparing TE score of each country. 

Introduction: Data used in this section should be updated to most recent years. Organization of remaining of this study should be outlined at the end of this section.

-We are so sorry for not updated the data, this is due to the availability of data in 2022 still have not been publish completely until today by several companies. If we try to add the research timeline until 2022, it will decrease number of research sample. Hope you can consider this part. 

-We add the organization of the article at the end of introduction part. 

Literature Review: This section should be divided into three parts. The first part must be based on the theoretical background. Secondly, the discussion should be developed on the estimation of technical efficiency. DEA and SFA approaches are used to estimate technical efficiency. This study employed the SFA approach to calculate the technical efficiency of halal food and beverages. Please justify the SFA approach compared to the DEA approach. Consult the latest published articles for your guidance and improve this section (https://doi.org/10.3390/en14133923;
https://doi.org/10.1007/s10668-020-01023-2;
https://doi.org/10.3389/fenrg.2022.943771;
https://doi.org/10.1007/s10668-022-02194-w). Third, include empirical literature that explores the impact of efficiency on firm value. Finally, find the research gap that this study fills.

We rewrite this sections following your comment. We separated the literature review into 3 part and close it with hypothesis. The theoretical background that we used is signaling theory, then for technical efficiency section, we emphasize the differences of DEA and SFA as well as the reason for choosing SFA approach for two steps estimation model. We also try to re-write the empirical review which support this research and find the research gaps which are lies in several factors.

Data and Statistical Method: (1). the title of this section should be “Methodology”. (2). It is recommended to include a table regarding “description of the data series” used in this study. (3). It is recommended to justify the significance of each variable included in the model. (4). Technical efficiency is estimated from the production function including capital, labor and other factors of the firm. This study is missing labor of the firm.

Thanks for the recommendation. 

- We re-new the section by “Methodology”. 

- We also explain the data series in Table 1 and describe every variable that has been used after the equation as well as connecting the reason for choosing the variables based on previous study. 

- We add the significant of each variable (at 5%) in every sub section of variables. 

- For labor variable, we are apologize for not including in our model due to several reasons: (1) The labor cost already include in liability variable, if we input again it will cause double counting; (2) we already try to separate between liability and labor cost, but the panel model shows the insignificant of TE partially and simultaneously as well as low R-square for each model. 

Results and Discussion: first, it is recommended to include descriptive statistics of the variables for each country used in the study in tabular form. This study is based on panel data with five cross sections and time period 2017-2021. Use the panel data SFA model to calculate the technical efficiency of each country with different time periods. It is recommended to use the above-mentioned articles for your guidance and to improve this section. Second, report findings of the SFA panel data model along with diagnostic statistics. Third, report each country's technical efficiency along with their ranking. Finally, to explore the impact of technical efficiency and other control variables on firm performance, use panel data models such as fixed effect, random effect, ARDL panel, or fully modified dynamic panel model, whichever best fits your data series. It is recommended to merge the discussion section with the "Results and Discussion" section.

Thanks for the suggestions

-We add the statistic descriptive table for SFA and Panel

-We cite the papers that you suggest in literature review and methodology parts.

-We explain the diagnostic statistic of each country to explain the reason of panel best model that we used.

-We rank the country from the highest to the lowest of TE score, which can be seen at table 8

- We edit the result and discussion section. At result, we explain the number, significances, and the impact of significant variable. Whereas, in discussion we try to collaborate the result with actual condition, grand theory that we used, as well as the previous study that support our finding. 

Conclusion: This section should be titled “Conclusions and policy Implications”. Include policy implications based on study results. It is recommended to include “Future Research Directions” at the end of this section.

-we rename this part as Conclusion and policy implications. At first paragraph we briefly explain the aim of study and its result, than following by explain the panel data approach result and suggestion. The third paragraph we emphasize the research implication for several parties (Firm, investor, and policy makers. In the end we provide the research limitation which also can be used as further direction for researcher.

---

## [Decision Letter · Decision Letter 2]

22 May 2023

THE IMPACT OF TECHNICAL EFFICIENCY ON THE FIRMS’ VALUE: THE CASE OF THE HALAL FOOD AND BEVERAGE INDUSTRY IN SELECTED COUNTRIES

PONE-D-22-31308R2

Dear Dr. Siti Zulaikha,

We’re pleased to inform you that your manuscript has been judged scientifically suitable for publication and will be formally accepted for publication once it meets all outstanding technical requirements.

With Kind Regards,

Asst. Prof. Dr. Nemer Badwan

PhD in Economics and Finance

Assistant Professor of Economics and Finance

Academic Editor

PLOS ONE

Additional Editor Comments (optional):

Reviewers' comments:

Reviewer's Responses to Questions

**Comments to the Author**

1. If the authors have adequately addressed your comments raised in a previous round of review and you feel that this manuscript is now acceptable for publication, you may indicate that here to bypass the “Comments to the Author” section, enter your conflict of interest statement in the “Confidential to Editor” section, and submit your "Accept" recommendation.

Reviewer #3: All comments have been addressed

Reviewer #4: All comments have been addressed

2. Is the manuscript technically sound, and do the data support the conclusions?

Reviewer #3: Yes

Reviewer #4: Yes

3. Has the statistical analysis been performed appropriately and rigorously? 

Reviewer #3: Yes

Reviewer #4: Yes

4. Have the authors made all data underlying the findings in their manuscript fully available?

Reviewer #3: Yes

Reviewer #4: Yes

5. Is the manuscript presented in an intelligible fashion and written in standard English?

Reviewer #3: Yes

Reviewer #4: Yes

6. Review Comments to the Author

Reviewer #3: Dear Authors, thank you for your modifications. I can suggest the acceptance of your research paper.

Best regards

The Reviewer

Reviewer #4: I appreciated your efforts to address my comments and suggestions. I recommend the manuscript for publication.

7. PLOS authors have the option to publish the peer review history of their article (what does this mean?). If published, this will include your full peer review and any attached files.

Reviewer #3: No

Reviewer #4: **Yes: **Prof. Dr. Dilawar Khan

<quillbot-extension-portal></quillbot-extension-portal>

---

## [Editor Report · Acceptance letter]

10 Jul 2023

PONE-D-22-31308R2 

The impact of technical efficiency on  firms’ value: the case of the halal food and beverage industry in selected countries 

Dear Dr. Zulaikha:

I'm pleased to inform you that your manuscript has been deemed suitable for publication in PLOS ONE. Congratulations! Your manuscript is now with our production department. 

Kind regards, 

on behalf of

Asst. Prof. Dr. Nemer Badwan 

Academic Editor

PLOS ONE